# Mortality Levels and Production Indicators for Suspicion of Highly Pathogenic Avian Influenza Virus Infection in Commercially Farmed Ducks

**DOI:** 10.3390/pathogens10111498

**Published:** 2021-11-17

**Authors:** Armin R. W. Elbers, José L. Gonzales

**Affiliations:** Department of Epidemiology, Bioinformatics and Animal Models, Wageningen Bioveterinary Research, 8221 RA Lelystad, The Netherlands; jose.gonzales@wur.nl

**Keywords:** breeder and broiler ducks, HPAI, reporting thresholds, early detection

## Abstract

(1) Background: Highly pathogenic avian influenza (HPAI) is a viral infection characterized by inducing severe disease and high levels of mortality in gallinaceous poultry. Increased mortality, drop in egg production or decreased feed or water intake are used as indicators for notification of suspicions of HPAI outbreaks. However, infections in commercial duck flocks may result in mild disease with low mortality levels, thereby compromising notifications. (2) Methods: Data on daily mortality, egg production, feed intake and water intake from broiler and breeder duck flocks not infected (*n* = 56 and *n* = 11, respectively) and infected with HPAIV (*n* = 13, *n* = 4) were used for analyses. Data from negative flocks were used to assess the baseline (daily) levels of mortality and production parameters and to identify potential threshold values for triggering suspicions of HPAI infections and assess the specificity (Sp) of these thresholds. Data from infected flocks were used to assess the effect of infection on daily mortality and production and to evaluate the sensitivity (Se) of the thresholds for early detection of outbreaks. (3) Results: For broiler flocks, daily mortality > 0.3% (after the first week of production) or using a regression model for aberration detection would indicate infection with Se and Sp higher than 80%. Drops in mean daily feed or water intake larger than 7 g or 14 mL (after the first week of production), respectively, are sensitive indicators of infection but have poor Sp. For breeders, mortality thresholds are poor indicators of infection (low Se and Sp). However, a consecutive drop in egg production larger than 9% is an effective indicator of a HPAI outbreak. For both broiler and breeder duck flocks, cumulative average methods were also assessed, which had high Se but generated many false alarms (poor Sp). (4) Conclusions: The identified reporting thresholds can be used to update legislation and provide guidelines to farmers and veterinarians to notify suspicions of HPAI outbreaks in commercial duck flocks.

## 1. Introduction

The almost annual introduction and spread of highly pathogenic avian influenza virus (HPAIV) of subtype H5Nx in parts of Asia, Europe, the Middle East, Africa and North America in the last two decades pose a potential threat for the global poultry industry as well as for human health. In response, countries and international organizations are closely observing this threat situation, assessing the possible risks for animal and public health, developing and implementing early detection systems [1,2,3]. HPAIVs that cause major diseases in poultry belong to the H5 and H7 subtypes, although not all viruses of these subtypes are HPAI viruses. The influenza A viruses of the remaining subtypes belong to the group of low pathogenic avian influenza (LPAI) viruses [4]. Chickens and turkeys infected with HPAIV are mostly found dead with clinical signs such as apathy, depression, reduction in normal vocalization, decreased feed and water consumption, and an exponential increase in mortality [1]. HPAIV infections of domestic bird species such as ducks and geese usually do not cause severe disease. Mild clinical signs and low mortality levels induced by HPAIV infection in farmed ducks pose an increased risk of late diagnosis or misdiagnosis, leading to delayed notification. However, non-specific clinical signs and above-baseline mortality in farmed ducks have been observed in recent HPAI outbreaks in Hungary, Italy, UK, South Korea and the USA [5,6,7,8,9,10]. 

Official requirements concerning the reporting—to the competent veterinary authorities—of a suspicion of a possible avian influenza virus (AIV) infection in poultry are stipulated in the Dutch Statutory Regulation “Prevention, eradication, and monitoring of contagious animal diseases, zoonoses and TSE’s” within the Dutch Animal Health and Welfare Law [11]. Requirements for reporting specific mortality thresholds to the competent veterinary authorities are stipulated for chicken-layers, chicken and broiler breeders and turkeys, but not for farmed ducks. There is a reporting threshold for poultry in general, which includes farmed ducks, of 3% mortality per week, but this has led to considerable delayed notification of clinically suspect situations and should be averted [12]. During the large 2003 HPAI subtype H7N7 epidemic in the Netherlands, only two duck farms became infected, with layers and turkeys making up the majority of infected species [1]. However, during the HPAI H5Nx epidemics in 2014, 2016, and 2017/2018, several commercial farmed duck flocks became infected in the Netherlands [13,14,15], and large epidemics affecting duck farms were observed in France in the periods 2016–2017 and 2020–2021 [16,17]. Therefore, there is an urgent need to develop effective reporting thresholds for suspicions of HPAI outbreaks in commercially farmed ducks in order to prevent delayed detection.

## 2. Results

### 2.1. Broiler Ducks

#### 2.1.1. Daily (Baseline) Mortality and Feed and Water Intake in Non-HPAIV-Infected Flocks

The expected daily baseline mortality in Dutch broiler duck (*Anas platyrhynchos domestica*) flocks during the production cycle was estimated using a generalized linear mixed model (GLMM) with a negative binomial distribution. The model parameters and results are shown in Appendix A and Appendix A, respectively. Mortality is expected to be the highest during the first week of life, the expected mean (95% confidence Intervals (CI)) mortality being 0.18% (0.04–0.52%). Mortality is the lowest between 7 and 37 (one week before slaughter) days of age. The expected mortality in this period was 0.04% (95% CI: 0.01–0.12%). During the last week of production (>37 days of age), mortality increases again, the expected mortality being 0.08% (95% CI: 0.03–0.20%). The expected standard deviation (SD) of the daily mortality was 0.006% (range: 0.003–0.008%) (approximately 1 in 10,000 broiler ducks). 

Feed and water intake were assessed by fitting a linear mixed model (LMM) (Appendix A). The mean increase in feed intake was estimated to be 5.8 (95% CI: 5.4–6.2) g/duck/day. The expected (variation) standard deviation in this daily increase was 3.4 (range: 2.5–6) g. The mean increase in water intake was 13.8 (95% CI: 13.0–14.5) mL/duck/day. The expected SD in daily water intake was estimated to be 6.9 (range: 5.5–11.6) mL. 

#### 2.1.2. Mortality and Feed and Water Intake in HPAIV-Infected Broiler Flocks

The observed mortality in the HPAIV-infected broiler duck (*A. platyrhynchos domestica*) flocks analyzed is shown in Figure 1. A different behavior in the daily increase in mortality can be observed among flocks affected with different HPAIV subtypes or similar subtype but different years. In all these outbreaks, increased mortality could be observed within the period of time with available data for analysis. 

Feed or water intake data were only available for six of these flocks and generally, different from mortality data, data for the previous one to four days before detection and culling were not available because the poultry farmer did not register them anymore (loss of focus due to the clinical situation) (Figure 2). Marked decreases in feed and water intake were only observed in flocks where an exponential increase in mortality could be observed and followed for more than 3 days before culling (2020 H5N8, flock 12 and 13) and there was also sufficient feed and water intake data (Figure 2).

#### 2.1.3. Detection Thresholds

The 95% upper confidence limit (UCL) of the expected daily mortalities during the production period were used to set and assess three fixed thresholds: 0.1%, 0.17% and 0.3% mortality. The first threshold (Q1) was similar to using the mean overall daily mortality + 2 * SD (0.08%). The performance results of these fixed thresholds and those of the flock-tailored detection methods (GLMM and moving average (CUSUM) are shown in Table 1. Graphical representations for some of these thresholds are shown in Figure 1 and Appendix A.

If an alarm is raised on the day a threshold is passed, all detection methods appear to be equally sensitive; however, if an alarm is raised when mortality passes the thresholds for two consecutive days (referred to as two-day-alarm), flock-tailored methods appear to be more sensitive. In addition, it appears that flock-tailored methods may also detect outbreaks earlier (Figure 1). As for the specificity, the CUSUM methods and fixed threshold of 0.1% mortality showed poor specificities (<0.5). The methods with the highest Sp which still showed good Se were the use of the GLMM as predictive tool or using a 0.3% fixed mortality threshold. In Figure 1, the application of these two thresholds in the HPAI-infected flocks is shown.

Assessed thresholds for mean daily feed intake per duck were drops of 7 or 9 g. As for water intake, drops of 14 and 18 mL in the mean daily water intake per duck were assessed. For both parameters, regardless of the threshold used, several false-positive signals per flock were generated (Appendix A). When these thresholds were applied to the infected flocks, “true” alarms were raised for those flocks (2020 H5N8, flocks 12 and 13) where enough data were available (Figure 2). In these flocks, the alarms based on these production parameters were raised one to two days before the mortality alarm. 

### 2.2. Duck Breeders

#### 2.2.1. Daily (Baseline) Mortality and Egg Production in Non-HPAIV-Infected Flocks

The model parameters and results are shown in Appendix A and Appendix A. The model-estimated mean daily mortality (females and males) during the production period was 0.05% (95% CI: 0.01–0.11%), with the expected SD equal to 0.010% (range: 0.006–0.016) (daily variation of approximately one dead breeder per 10,000 ducks in a flock). No significant differences in daily mortality were observed between female and male breeders. 

Daily egg production and the model fits are shown in Appendix A. Peak production in the flocks was reached when breeders were on average 251 days old (range: 213–281 days). Egg production reached an average peak of 94.5% (range: 87.3–98.2%). Given the daily decreasing trend in production following peak of production, we were mostly interested in estimating the daily variation in egg production. The model-estimated expected daily SD of the egg production was ±4.59% (range: 4.50–4.61%). 

#### 2.2.2. Mortality and Egg Production in HPAIV-Infected Flocks and Performance of Detection Thresholds

A consistent increase in daily mortality was observed only in one out of the four affected flocks (flock labelled as “H5N8 2014”) (Figure 3). Mortality in the other flocks did not show continuous daily increases as observed for example in some of the affected broiler flocks (Figure 1). In contrast to the observations with respect to mortality, large and continuous drops in egg production were observed in all these flocks (Figure 3), with egg production dropping from levels approximately 75% to lower than 25% depending on the time of HPAIV introduction. For flock “H5N8 2017 1C”, egg production was as low as 5.9% when detection of infection took place.

The model that best explained the drop in egg production was a GLMM with a binomial distribution (Appendix A). The average odds for a decrease in daily egg production when compared to the “baseline production” was estimated to be 0.37 (95% CI: 0.36–0.38) per day, which when converted to probability, say for the first 5 days (to cover the flock with the longest number of observations before detection “H5N8 2017 1C”), translates into drops in production, when compared to the previous day, ranging from 9% to 24%. With 9% being the difference in production at day five (last day) of the decreasing period. 

The 95% UCL of the expected daily mortalities during the production period were used to set three fixed thresholds: 0.07% (mean + 1.96 SD), 0.2% (Q1) and 0.3% (Q2) mortality. The 0.07% threshold showed very poor Sp, whilst the 0.2% and 0.3% thresholds showed better Sp when using a two-consecutive-days alarm (Table 2, Appendix A). Given the small number of infected farms no formal assessment of Se was done. If the 0.2% or 0.3% thresholds were being applied, these would be able to detect outbreaks where mortality behaves similarly to the H5N8 2014 affected flock. 

As for egg production, we used a drop in production equal or greater than 9% (1.96 * SD) as the threshold in relation to the production of the previous day. In Figure 3, it can be seen that this threshold raised alarms daily during the outbreak period in the infected flocks. This alarm was not raised on the last day of detection of flock “H5N8 2017 1C”. This confirms the binomial model predictions that drops in egg production (compared to the previous day) would be approximately 9% (see above) following a period of five days decrease in egg production. This threshold also shows good Sp (Table 2, Appendix A). When we used this threshold to assess the drop in egg production in relation the mean production the previous week, a similar performance was observed (Appendix A).

We also tried the CUSUM and the weekly ratio methods for daily mortality and egg production, but these methods showed very poor Sp (data not shown).

## 3. Discussion

Based on daily mortality (for broiler duck flocks) and daily egg production (for breeder duck flocks), effective reporting thresholds were identified with high Se and Sp. Finding an effective mortality threshold for broiler duck flocks was complicated, since the course of baseline mortality of a non-HPAIV-infected flock over the short production period for a given broiler duck flock shows a bimodular or U-shaped distribution: (i) increased mortality in the first week of production; (ii) relatively low mortality in the production period between 7 and 37 days after start of production; (iii) increased mortality in the week before slaughter. Increased first-week mortality (FWM) and mortality in the week before slaughter is a common phenomenon in the broiler industry, not only for ducks but also for broiler chickens [18,19,20]. The first week of life of ducklings or chicks is a delicate period where several physiological systems and organs are still in an immature phase [21]. It is a transition period from a controlled hatchery environment to a more autonomous life at the farm, which requires adaptation to changes. Housing factors and management routines, both at the hatchery and at the broiler farm, are associated with FWM [20,22]. If the young birds fail to adapt to the changes and challenges they meet after arrival at the broiler farm, this could result in increased FWM. Increased mortality in broilers in the week before slaughter is likely due to health problems such as heart attack, ascites and leg problems [19]. In order to circumvent the common FWM problem, the assessed mortality thresholds for broiler duck flocks should be applied in practice after the first 7 days of production. 

For reporting thresholds for early detection of HPAI outbreaks to be effective, the likely benefit of early detection of an outbreak (true alarm) should be balanced by the economic and social costs associated with a rapid response triggered by a false alarm [2]. An increase in mortality after infection with HPAIV is not pathognomonic; it can be caused by a range of other poultry diseases or management-related problems such as failure of ventilation and heating and will lower the Sp of the reporting threshold. 

For broiler duck flocks, a fixed 0.3% mortality threshold after the first week of production is an effective reporting threshold, combining high Se with high Sp; the associated mean false-alarm rate of 9 per 1000 alarm signals is in our view socially acceptable. Assessed reporting thresholds for mean daily feed intake (decrease in mean daily feed intake per duck ≥ 7 g) and water intake (decrease in mean daily water intake per duck ≥ 14 mL) showed a high Se, and the alarms based on these production parameters were raised even one to two days before the mortality alarm signal in HPAIV-infected farms. Unfortunately, Sp of these thresholds are poor. The take-home message for this would be that if the above-mentioned decrease in mean daily feed intake and/or water intake is observed in a broiler duck flock, one should always consider submitting blood and swab samples for diagnostic testing to exclude avian influenza virus as the cause of the observed deviation in food and/or water intake [23].

Based on the limited data available for breeder duck farms, daily mortality was not a consistent indicator for a suspect HPAIV infection in the flocks; a consistent increase in daily mortality was observed only in one out of the four affected flocks. This can be due to the fact that we only had access to four HPAIV-infected duck breeder flocks as basis for the analysis; but it can also be that this reflects the true situation in the field. More data from other HPAIV outbreaks in breeder duck flocks from other countries would be helpful to get more insight into this phenomenon. Nevertheless, a 0.3% mortality threshold should not be ignored when increased mortality is observed in breeder duck flocks. Fortunately, a decrease in egg production of >9% in relation to the production of the previous day serves as an excellent reporting threshold (high Se and Sp) for detection of HPAIV-infected breeder duck flocks. 

There are a few limitations to consider in our study. Unfortunately, only a few farms kept record of food and water intake, although the potential of those data as indicators for health deviations in poultry is without saying [12]. With that, it was difficult to use these parameters for the development of detection thresholds. Furthermore, we only had data from a limited number of HPAIV outbreaks to assess the sensitivity of the reporting thresholds. With more outbreak data, we probably would have been able to compare possible thresholds with even higher precision. With respect to the not-infected “healthy” duck flocks, there was not much information and/or confirmed causes in the data itself with respect to incidental increased mortality which has led to some false positive alarms. There were a few exceptions with documented information in the production calendars which led to exclusion of the data for analysis like, e.g., increased mortality due to infection with *Riemerella anatipestifer*. 

As for the flock-tailored methods assessed in this study, the CUSUM methods, even though they had high Se, had poor Sp, resulting in high number of false alarms. Similar results for this type of method were also observed when they were applied to trigger suspicions of HPAI outbreaks in layer chickens [2]. The application of the GLMM predictions (upper confidence limits) as daily threshold tailored to the age of the duck performed well. A similar approach has also been previously assessed for layer chickens [24]. Ideally, such an approach could be implemented at the flock level, using historical data from previous production rounds (flocks) in the farm to predict the expected mortality for every new flock, with data from each production round being used to update the GLMM. This continuous updating based on the specific farm performance would contribute to continued optimization of the efficacy of the system. 

## 4. Materials and Methods

### 4.1. Data

#### 4.1.1. Non-HPAIV-Infected Flocks

Data from broiler duck flocks and breeder duck flocks which tested negative in the Dutch serological avian influenza surveillance system [25] was obtained for this analysis. A flock was defined as a group of ducks starting a production cycle at the same date in a poultry house. For each flock, records of the starting date of production cycle, total number of ducks starting production, daily mortality, egg production (for duck breeder farms), feed and water intake (if available) were provided by the largest Duck Integration in the Netherlands. Data from 56 broiler duck flocks, originating from 15 Dutch duck broiler farms (there are a total of 45 duck broiler farms in the Netherlands) were provided. Three out of these 56 flocks were excluded from analysis because these flocks showed periods of increased mortality (data not shown) within their production cycle due to a specific bacterial infection (*Riemerella anatipestifer*). The average duration of a production cycle of a flock included in the analysis was 44 days (standard deviation (s.d.): 2.5 days). 

A total of 11 breeder duck flocks were included originating from two Dutch breeder farms (there are a total of 11 duck breeder farms in the Netherlands). The average duration of a production period of a duck breeder flock was 53 weeks (s.d.: 3.9 weeks). Each production period started when the average age of the breeder ducks was 19 weeks (s.d.: 1.9 weeks) and ended when the average age of the breeder ducks was 72 weeks (s.d.: 2.7 weeks). The data were provided in the form of production calendars (on paper) with manual daily recording of mortality, production parameters and veterinary treatments or other peculiarities. Production calendar data on paper were digitalized and transferred to Microsoft Excel sheets.

#### 4.1.2. HPAIV-Infected Duck Flocks

Mortality and production data from confirmed HPAIV-infected duck flocks (13 broiler duck flocks and 4 breeder duck flocks) were obtained from: (a) the Netherlands’ Food and Consumer Product Safety Authority (broiler duck flocks infected with HPAIV subtype H5N8 and subtype H5N6 in 2014, 2016, 2017, 2018 and 2020); (b) the Friedrich Loeffler Institute in Germany (a broiler duck flock infected with HPAIV subtype H5N1 in 2007 and a broiler duck flock infected with HPAIV subtype H5N8 in 2017; three breeder duck flocks (one farm, three poultry houses) infected with HPAIV subtype H5N8 in 2017); (c) an outbreak described in the United Kingdom in 2014 (one breeder duck flock infected with HPAIV subtype H5N8) [8]; (d) an outbreak described in the United States of America in 2015 (one broiler duck flock infected with HPAIV subtype H5N8) [9]. 

### 4.2. Production Parameters and Detection Algorithms

#### 4.2.1. Daily Mortality 

Similar methods to those developed for chicken layers [2] were also applied for this study. In short, these methods were based on (i) the assessment (determination) of the expected (baseline) daily mortality observed during a “normal” production cycle, which leads to defining fixed reporting thresholds (triggers) and (ii) flock-tailored triggers based on moving average thresholds or flock-tailored prediction of expected daily mortality. 

Fixed Mortality Thresholds

Daily mortality rates in non-HPAIV-infected duck flocks were modelled using generalized linear mixed models (GLMM) with either a Poisson or a negative binomial distribution (to account for overdispersion observed when fitting models with a Poisson distribution), where the daily number of dead ducks was the explanatory variable; the natural log of the daily population size of the flock was the offset, and the age of the ducks in days was the response variable. To account for deviations in linearity in time, natural cubic splines were used on the variable age. The flock identifier was used as the grouping variable (random effect). The models were used to calculate the daily expected mortality and variance (which combines the variance of the fixed and random variables) in a non-infected flock and corresponding 95% upper confidence limits (UCL), estimated by bootstrapping. The latter were considered the maximum mortality rates expected in a non-infected flock and were used to set fixed reporting thresholds. The first (Q1), second (Q2) and third quantiles (Q3) of the estimated UCL were selected and evaluated as reporting thresholds. 

Flock-Tailored Triggers

Two approaches were assessed: a) application of the developed GLMM to predict the expected daily (baseline) mortality and 95% UCL and use of this prediction to identify increases in mortality outside the expected baseline mortality; b) use of a moving average (CUSUM) method which uses data on the observed mortality in previous (defined number of) days to inform the expected (baseline) mortality in a flock at the day of evaluation, which depending on the algorithm a one- (EARS-1) or two-day (EARS-2) time lag is used between the baseline mean and the day of evaluation [26]. For broiler ducks, a three or a five days moving average, to determine the baseline mortality, was assessed. As for breeder ducks, a seven-day moving average was used. Detailed descriptions of the algorithms used for the CUSUM method are provided elsewhere [2,26].

#### 4.2.2. Feed and Water Intake

This information was only available for 38 non-HPAIV-infected duck broiler flocks. The expected mean and variance of the daily rate of increase in either feed intake (gr of feed/duck/day) or water intake (ml water/duck/day) per duck during the production period were estimated fitting liner mixed models (LMM). The models that best fitted that data had feed or water intake as the response variables, age in days was the explanatory variable (fixed effect), the flock id was introduced as a random intercept (grouping variable) and age in days as random slope within each flock. The estimated variance was the sum of the fixed and random variance of the model predictions. To set reporting thresholds, 1.96 or 2.6 times the expected daily standard deviation (SD = square root of variance) were used as thresholds for the expected maximum drop in feed or water intake.

#### 4.2.3. Egg Production

The expected (mean and variance) daily egg production, expressed in percent production (total eggs per day/total number of ducks in production * 100), during the production cycle was estimated by fitting a LMM where percent egg production was the response variable, the ducks’ age in days was the explanatory variable, which included natural cubic splines to account from deviations from linearity in time, and the flock identifier was used as grouping variable (random effect). This model was fitted to data from the non-HPAIV-infected flocks. To set reporting thresholds, similar approaches as those used to set thresholds for mortality were tried. Additionally, 1.96 * SD was used as an indicator of the expected maximum drop in (percent) egg production. This drop was assessed either using as reference the observed egg production the previous day or the mean egg production of the previous week. 

In addition, the level or rate of decrease in egg production observed in HPAIV-infected flocks was estimated by selecting the days of observed decrease assumed to be due to HPAIV (Figure 2b) and fitting a LMM (percent egg production) or GLMM with a binomial distribution (proportion egg production) to the observed daily egg production.

### 4.3. Evaluation of Performance

For this evaluation, an “alarm signal” was defined as an increase in mortality (either for one or two consecutive days) above the mortality or production thresholds set by the detection method. A “true alarm” would be an alarm raised any time during a confirmed HPAI outbreak. Alarms outside an assumed outbreak period or raised in non-infected farms were considered “false alarms” for detection of HPAI. 

Parameters used to evaluate the performance (accuracy) of the detection methods were sensitivity (Se), timeliness (T), specificity (Sp) and the false-positive rate (FP). The first two parameters were evaluated applying the detection algorithms on HPAIV-infected flocks and Sp was evaluated applying the algorithms on non-infected flocks. Se was defined as the ability of the reporting method to trigger a “true” alarm in an actively infected flock, any time during the course of the outbreak (Se = number of detected infected flocks/total number of infected flocks). T is a measure of how early the detection method triggered an alarm in relation to the actual detection day of the outbreak. Sp was defined as the ability of the algorithms to not trigger alarms (“false alarms”) in non-infected flocks during the whole production cycle (Sp = number of flock with no alarm/total number of non-infected flocks). FP was defined as the probability of FP alarms per day (FP = number of false alarms/total observation days).

### 4.4. Data Analysis Software

All data analysis was performed using the statistical software package R version 4.0.2 [27]. The LMM and GLMM were fit using the library lme4 [28] and the model’s prediction of fitted values and bootstrap estimation of confidence intervals was done using the library merTools [29]. Figures were designed using the library ggplot2 in R [27]. 

## 5. Conclusions

Effective thresholds for reporting suspicions of HPAIV outbreaks in commercial broiler and breeder duck flocks were identified. It is clear that the existing reporting threshold in the acting Statutory Regulation (≥3% mortality per week) for commercial duck flocks needs to be updated and we provide new thresholds for early detection of HPAIV outbreaks on commercial duck farms. The identified reporting thresholds can be used to update existing legislation, inform policy makers and provide guidelines to farmers and veterinarians to notify suspicions of HPAIV outbreaks in commercial duck flocks.

## Figures and Tables

**Figure 1 pathogens-10-01498-f001:**
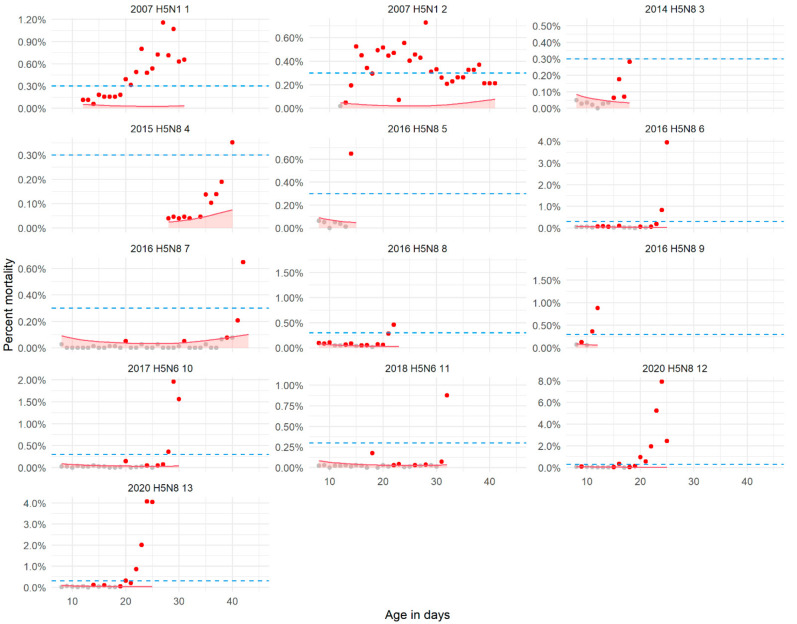
Example of the application of the developed generalized linear mixed model (GLMM) and the fixed threshold (mortality = 0.3%) for raising alarms during outbreaks of highly pathogenic avian influenza (HPAI). The GLMM predicts the expected daily (baseline) mortality, based on the age of the flock, in the absence of infection (last dot in each flock figure is the day of detection). The red line represents the expected 95% upper confidence limit (UCL threshold) of the baseline mortality. Red dots are raised alarms following mortality levels higher than the UCL threshold (one day alarm). Blue dotted line indicates the fixed threshold. Notice that using this fixed threshold, alarms would be raised with less frequency and later than using the GLMM-UCL.

**Figure 2 pathogens-10-01498-f002:**
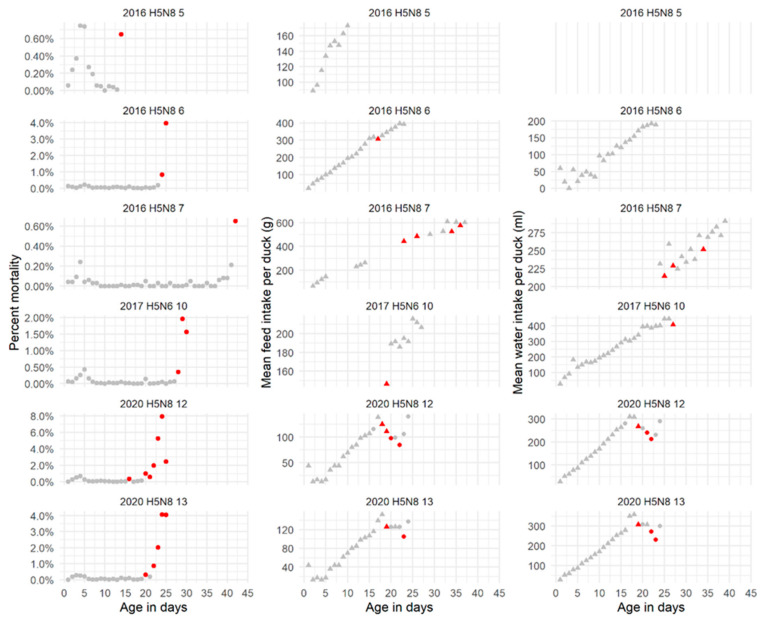
Daily mortality (**left**) and mean daily feed (**middle**) and water (**right**) intake observed in duck (*Anas platyrhynchos domestica*) broiler flocks up to the time of detection of a highly pathogenic avian influenza infection (there was no data available on water intake for case 2016 H5N8 5, right upper corner. Dots and triangles in grey are daily observations; dots and triangles marked in red indicate signals generated based on the corresponding identified threshold: daily mortality >0.3%, drop in mean daily feed intake larger than 7 g/duck/day, drop in mean daily water intake larger than 14 mL/duck/day.

**Figure 3 pathogens-10-01498-f003:**
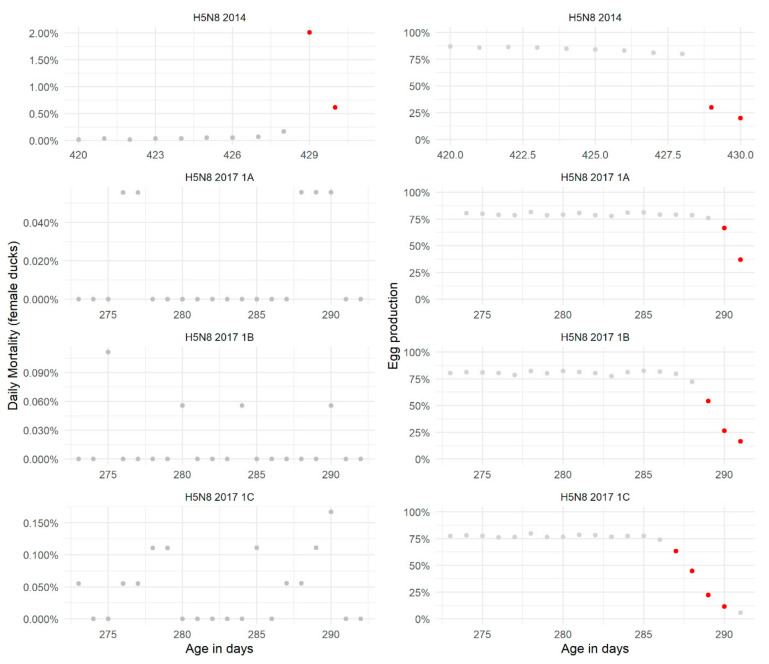
Observed daily mortality (**left**) and egg production (**right**) in breeder duck (*Anas platyrhynchos domestica*) flocks infected with highly pathogenic avian influenza virus (HPAIV). The virus subtype and year of the outbreak are given as titles for each panel. For flocks affected with the same HPAIV subtype the same year, a flock identifier (1A, 1B, 1C) is provided. Dots in grey are daily observations; dots marked in red identify days when: the percent mortality was higher than 0.2% or 0.3%, the percent drop in egg production is larger than 9% of the egg production observed the previous day.

**Table 1 pathogens-10-01498-t001:** Performance of the reporting thresholds (methods) applied to monitor broiler duck (*Anas platyrhynchos domestica*) flocks. Performance is assessed considering an alarm would be raised on the day a threshold is passed or after two consecutive days.

Threshold	Sensitivity (*n* = 13)		Median (Range) Days Earlier Detection		Specificity (*n* = 53)		False Alarm Rate (95% Confidence Interval)	
	1 day	2 days	1 day	2 days	1 day	2 days	1 day	2 days
0.1%	1.0	0.769	2 (0–6)	1 (0–5)	0.245	0.491	0.104 (0.091–0.119)	0.066 (0.056–0.078)
0.17%	1.0	0.769	2 (0–7)	1 (0–4)	0.547	0.698	0.041 (0.033–0.051)	0.022 (0.016–0.029)
0.3%	0.923	0.538	1 (0–5)	1 (0–5)	0.811	0.962	0.009 (0.005–0.014)	0.001 (0.000–0.004)
GLMM ^a^	1.0	0.923	9 (3–11)	3 (2–11)	0.792	0.962	0.009 (0.006–0.014)	0.002 (0.001–0.004)
CUSUM-1 ^b^	1.0	1.0	5 (1–6)	3 (0–4)	0.038	0.170	0.216 (0.198–0.234)	0.109 (0.096–0.123)
CUSUM-2 ^c^	1.0	1.0	5 (1–6)	3 (0–4)	0.057	0.226	0.206 (0.189–0.225)	0.104 (0.091–0.119)

^a^ Generalized mixed linear model (GLMM). ^b^ Moving average (CUSUM-1) method using a one-day (EARS-1) time lag between the baseline mean and the day of evaluation. ^c^ Moving average (CUSUM-2) method using a two-day (EARS-2) time lag between the baseline mean and the day of evaluation.

**Table 2 pathogens-10-01498-t002:** Performance of the reporting thresholds (methods) applied to monitor breeder duck flocks.

Threshold	Specificity (*n* = 10)		False Alarm Rate (95% CI)	
	1 Day	2 Days	1 Day	2 Days
**Mortality**				
0.07%	0.000	0.000	0.2847 (0.2704–0.3000)	0.1142 (0.1044–0.1250)
0.2%	0.000	0.600	0.0348 (0.0294–0.0412)	0.0043 (0.0026–0.0070)
0.3%	0.400	0.900	0.0088 (0.0063–0.0124)	0.0005 (0.0001–0.0019)
**Drop in egg production**				
>9%	0.636	1.0	0.0016 (0.0007–0.0038)	0.0000 (0.0000–0.0012)

## Data Availability

Data used for this study were made available from the largest duck integration in the Netherlands (AIV non-infected production rounds) and the Netherlands’ Food and Consumer Product Safety Authority and the Friedrich Loeffler Institute in Germany (HPAI-infected duck farms). Restrictions to the availability of these data apply, the data were used under license and are not publicly available. Data are possibly available from the authors upon reasonable request and with permission of Tomassen Duck-to Farm B.V., the Netherlands’ Food and Consumer Product Safety Authority and the Friedrich Loeffler Institute, Germany.

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
