# Peer review of "Mortality Levels and Production Indicators for Suspicion of Highly Pathogenic Avian Influenza Virus Infection in Commercially Farmed Ducks"

_pathogens, 2021, doi:10.3390/pathogens10111498_

Round 1

Reviewer 1 Report

Summary

The manuscript by Elbers and Gonzales entitled, “Mortality levels and production indicators for suspicion of highly pathogenic avian influenza virus infection in commercially farmed ducks” describes a study on establishing thresholds of farm performance parameters such as daily mortality and feed and water intake that are used to raise suspicion of HPAIV infection. The investigators used data from known non-HPAIV infected flocks to establish baseline models for farm performance parameters and compared them to that of known HPAIV-infected flocks. Their findings suggest that current standards for farm parameter thresholds to raise suspicion of HPAIV infection should be revisited.

Questions for discussion

What were the species of ducks used? There are known differences in disease outcomes in HPAIV-infected ducks, depending on species (e.g. Anas platyrhynchos and Cairina moschata).

Can you control for the age of ducks at the time of infection? Age has also been shown to affect mortality and clinical disease, with younger birds typically being more susceptible to disease.

Specific comments

Line 13: “poultry” to “gallinaceous poultry”. To be clearer, the HPAI refers to disease outcomes in gallinaceous species such as chickens, turkeys, and quail. Domestic Anseriformes species can also be considered as poultry but typically do not have severe AIV infections (with exceptions of course).

Line 59: Please clarify which species are referred to by “layers and breeders”

Line 74: Please spell out the acronym “GLMM”

Line 83: Please spell out the acronym “LMM”

Line 99: “floc” to “flock”

For Figures 1-3: Please use the same scales on the y-axis for the plots to make them easily comparable to one another. You can use a broken axis if needed. Also please indicate what the number at the end of the plot titles are.

Figure 2: The plot for “2016 H5N8 5” does not have any points at all.

Author Response

Reviewer 1

What were the species of ducks used? There are known differences in disease outcomes in HPAIV-infected ducks, depending on species (e.g. Anas platyrhynchos and Cairina moschata).

Response by authors: this has been added to the text, was Anas platyrhynchos domestica, see lines 73, 91, 113 etc. in the revised manuscript

Can you control for the age of ducks at the time of infection? Age has also been shown to affect mortality and clinical disease, with younger birds typically being more susceptible to disease.

Response by authors: age of the ducks was controlled for in the modelling of time of infection.

Specific comments

Line 13: “poultry” to “gallinaceous poultry”. To be clearer, the HPAI refers to disease outcomes in gallinaceous species such as chickens, turkeys, and quail. Domestic Anseriformes species can also be considered as poultry but typically do not have severe AIV infections (with exceptions of course).

Response by authors: is changed according to request by reviewer, see line 13 in revised manuscript.

Line 59: Please clarify which species are referred to by “layers and breeders”

Response by authors: additional text is added, see line 59 in revised manuscript.

Line 74: Please spell out the acronym “GLMM”

Response by authors: additional text is added, see line 74 in revised manuscript.

Line 83: Please spell out the acronym “LMM”

Response by authors: additional text is added, see line 84 in revised manuscript.

Line 99: “floc” to “flock”

Response by authors: is changed according to request by reviewer, see line 99 in revised manuscript.

For Figures 1-3: Please use the same scales on the y-axis for the plots to make them easily comparable to one another. You can use a broken axis if needed. Also please indicate what the number at the end of the plot titles are.

Response by the authors: as suggested by the reviewer, we checked if the use of the same scales would make it more easy to compare the information in the figure. Unfortunately, that is not the case, because it turns out that only one plot within the figure is then nice, but because the other plots have much lower values, these plots then show no informative data in the plot. So, for that reason, we have good reasons to stick to the original set up of the figure. We also checked if it is possible to use broken axis, but this is not possible using the library ggplot2 in the statistical software package R, with which we designed the figures.

Figure 2: The plot for “2016 H5N8 5” does not have any points at all.

Response by the authors: we have added text to the legend of figure 2, see line 115 in revised manuscript.

Reviewer 2 Report

Excellent work. I think you are underestimating the results found in the study. It is true that flock tailored methods had poor specificity, but the higher sensitivity can still be useful. While the method might not be appropriate for reporting, it can be considered a trigger for testing. Combining the sensitive parameters like drop in feed or water with a highly specific laboratory testing method can yield a more effective ways of discovering infected flocks, which can lead to early reporting. You eluded to this on lines 235 – 239; however, I think this should be one of the major findings of this manuscript.

Author Response

Reviewer 2

Comments and Suggestions for Authors

Excellent work. I think you are underestimating the results found in the study. It is true that flock tailored methods had poor specificity, but the higher sensitivity can still be useful. While the method might not be appropriate for reporting, it can be considered a trigger for testing. Combining the sensitive parameters like drop in feed or water with a highly specific laboratory testing method can yield a more effective ways of discovering infected flocks, which can lead to early reporting. You eluded to this on lines 235 – 239; however, I think this should be one of the major findings of this manuscript.

Response by the authors: the flock tailored methods, with high sensitivity, had poor specificity, resulting in high number of false alarms. High number of false alarms will lower the willingness of poultry famers to notify a suspect situation, and this is not what we want. Secondly, the use of flock tailored methods is a theoretical possibility, but is far from practical application in the field. It would mean the use of historical data from previous production rounds (flocks) in the farm, to predict the expected mortality for every new flock, with data from each production round being used to update the GLMM. This continuous updating based on the specific farm performance would mean online data sharing between the farm and a device that would run the GLMM, which is far from possible right now in practice. So we keep the original text as outlined in lines 285-295.